# Spontaneous Pneumomediastinum in Children with Viral Infections: Report of Three Cases Related to Rhinovirus or Respiratory Syncytial Virus Infection

**DOI:** 10.3390/children9071040

**Published:** 2022-07-13

**Authors:** Johanna L. Leinert, Alba Perez Ortiz, Neysan Rafat

**Affiliations:** 1University Children’s Hospital, Medical Faculty Mannheim, Heidelberg University, 68167 Mannheim, Germany; johanna.leinert@umm.de; 2Department of Neonatology, Medical Faculty Mannheim, Heidelberg University, 68167 Mannheim, Germany; alba.perez-ortiz@umm.de

**Keywords:** pneumomediastinum, children, viral infection, rhinovirus infection

## Abstract

Background: Spontaneous pneumomediastinum (SP) is generally a benign condition which can have various etiologies. Data on SP related to respiratory viral infections in children are rare and there are currently no official guidelines or consistent treatment recommendations for these patients. Aim: To discuss treatment options considering the recommendations for SP with different etiologies. Methods: We report three cases of SP, which were related to rhinovirus or respiratory syncytial virus (RSV) infection. Results: All three patients presented with typical symptoms of a respiratory tract infection and required oxygen supplementation during the hospital stay. All children benefited from a conservative, supportive therapy, and bed rest, and could be discharged after seven days or less without remaining symptoms. Conclusion: Surveillance and monitoring might be reasonable to detect and treat potential complications in children with SP due to viral infections, as one child developed an increasing pneumothorax, which had to be treated with a thoracic drainage.

## 1. Introduction

A pneumomediastinum is generally defined as the presence of interstitial air in the mediastinum. A spontaneous pneumomediastinum (SP) is caused by an increased pressure in the airways in comparison to the surrounding tissue, leading to a spread of air into the mediastinum without any external cause, such as trauma or surgery. In this incidence, air is often also pressed into the subcutaneous tissue or the pleural space, which is why an SP is often accompanied with subcutaneous emphysema and pneumothoraces [1]. In most cases, an SP presents with mild symptoms of chest and neck pain, sore throat, or dyspnea [2]. In children, SPs occur most commonly in asthma exacerbation or after a Valsalva maneuver such as vomiting or coughing [3,4,5]. Other triggering factors can be bronchospasms or foreign body ingestions. Only 13% to 20% of SPs have been described as occurring related to viral infections [4,5]. Recently they have also been described for patients with COVID-19 infections [6,7,8].

SPs related to respiratory viral infections in children are rare and only a few case reports are published [9,10]. Studies focusing on SPs due to respiratory viral infections are not available. Accordingly, data regarding the clinical course in these patients is rare, and there are currently no official guidelines or consistent treatment recommendations for those patients.

In this work, we report three cases of SP, which were related to rhinovirus or respiratory syncytial virus (RSV) infection in children, and discuss treatment options for these patients, considering the recommendations for SPs with different etiologies.

## 2. Case Reports

### 2.1. Case 1

Patient 1 was a three-year-old girl presenting with fever and cough for two days as well as tachydyspnea and neck swelling. In the clinical examination subcutaneous emphysema was detected above the clavicles and bilateral rhonchus was auscultated. Accordingly, a chest X-ray was performed, which showed a pneumomediastinum and a large subcutaneous emphysema (Figure 1). A polymerase chain reaction (PCR) of the nasal secretions obtained in our emergency unit was positive for RSV. High C-reactive protein (CRP) levels (108 mg/L, normal range 0–5 mg/L) led to the assumption of bacterial superinfection, which led to a treatment with ampicillin (100 mg per kg body weight per day, divided in three doses) for seven days. Oxygen supplementation was necessary to keep the saturation between 88% and 92%. Afterwards, oxygen supplementation could be stopped as the saturation was normal. Additionally, she received supportive intravenous fluid, and bed rest was prescribed. She recovered and was discharged after seven days.

### 2.2. Case 2

Patient 2 was a three-year-old girl with a history of three days of fever, obstructive bronchitis, and dyspnea, as well as a cough with sputum. On physical examination she presented with bilateral wheezing, rattling noises, and palpable subcutaneous emphysema. Multiplex PCR of the nasopharyngeal aspirate revealed an acute rhinovirus infection. In addition, this patient was treated with antibiotics due to bacterial superinfection (also suspected because of elevated CRP levels (99 mg/L) and received intravenous fluids (1.2 L of saline per day (85 mL/kg/day)). Therapy with the broad-spectrum antibiotic agent ceftriaxone (75 mg per kg body weight per day) had already been started by the referring hospital. To treat the obstructive bronchitis, she received inhalation treatment with salbutamol and corticosteroids. She required oxygen supplementation via nasal canula to maintain an oxygen saturation > 94%. Chest radiograph showed a pneumomediastinum in combination with pneumothorax and subcutaneous emphysema (Figure 2A). Due to deterioration of symptoms a follow-up X-ray was performed and indicated an increase in the pneumothorax and pneumomediastinum. A subsequent computed tomography (CT) of the thorax showed the increasing pneumomediastinum and pneumothorax, as well as bilateral atelectases (Figure 2B,C). Because of pulmonary deterioration with increasing dyspnea, a thoracic drainage was applied to treat the pneumothorax, which led to improvement within the next days. The patient recovered, the drainage could be removed after three days, and she was discharged after seven days.

### 2.3. Case 3

Patient 3 was a six-and-a-half-year-old girl who was admitted to our hospital with dyspnea and epigastric and chest pain after a history of a respiratory infection during the days before. Because of inspiratory and expiratory stridor, inhalation with bronchodilators was established and oxygen supplementation was started. In the clinical examination, a massive subcutaneous emphysema of the neck and chest was observed. Blood tests detected elevated infection markers (leukocytes of 15.4/nL (normal range 4.7–10.3/nL) and CRP of 58 mg/L). Therefore, we suspected bacterial superinfection and started empiric antibiotic treatment with ampicillin (for dosage, see case 1). Multiplex PCR of the nasal secretion was positive for rhinovirus. An X-ray was performed, showing a pneumomediastinum and pneumothorax as well as a subcutaneous emphysema (Figure 3). Because of suspected pneumopericardium in the chest X-ray and due to aggravation of the patient’s symptoms, an additional chest CT was performed. The CT ruled out a pneumopericardium and confirmed the pneumomediastinum and pneumothorax, and additionally showed pneumorrhachis (presence of air in the spinal epidural space). We carefully observed the patient, prescribed bed rest, and performed supportive treatment with oxygen therapy and intravenous fluid. The patient improved within the next few days and an X-ray before discharge showed an improvement of the initial findings. A few days later the symptoms completely resolved, and the patient was discharged after six days.

## 3. Discussion

In this report, we investigated three cases of children with SP caused by viral infections of the airways who could be effectively managed with supportive therapy, rest, and monitoring.

RSV and rhinovirus are two of the most common pathogens causing viral respiratory infections in children and are often seen in winter and beginning of spring [11]. Children under the age of two years, especially prematurely born toddlers, and those with chronic illnesses, such as immunodeficiencies or broncho-pulmonary disease, are often hospitalized with RSV [11]. The coincidence of RSV infection and SP has only been described in two other case reports before, and only one case of coincident rhinovirus infection and SP has been described [9,10]. The typical symptoms of these infections, dyspnea and coughing, which were present in all of our patients, can lead to an increased intrathoracic pressure, resulting in alveolar rupture, leading to SP.

In children presenting with dyspnea, chest pain, or subcutaneous emphysema, a chest X-ray may be indicated, which normally leads to the diagnosis of SP, if present. In unclear cases or suspected complications, such as pneumopericardium and tension pneumothorax or to exclude an esophageal rupture, a CT scan can be performed, as was the case with patients 2 and 3 of our report. In patient 3, we also observed a pneumorrhachis, which may appear to be a serious complication. However, previous studies showed that pneumorrhachis is self-limiting and does not require specific treatment in the vast majority of cases [12].

All three of our patients received supportive therapy with oxygen via nasal cannula and intravenous fluids which led to a substantial improvement of their symptoms. In all cases, blood tests showed increased inflammatory markers (leucocytes and CRP), which lead to the suspicion of bacterial superinfection. Therefore, all children received empiric intravenous antibiotic treatment. Only patient 2 needed additional invasive treatment in terms of a thoracic drainage to treat an enlarging pneumothorax which caused increasing dyspnea.

There are currently no guidelines and only a few treatment recommendations for children with SP. This might contribute to the fact that SP is often overinvestigated and overtreated [9,12,13]. Our cases suggest that SP in children with viral infections can be effectively treated with the above-mentioned supportive therapies.

A few larger studies investigated children with SP; however, in these studies the underlying cause for SP was manifold and their conclusions and recommendations were not restricted to or focused on viral infections [4,5,14]. In addition, two review articles were published, summarizing the diagnosis and management of children with SP; however, these also did not specifically address the etiology of the disease [1,15]. These studies and reviews recommend a conservative, non-invasive treatment for pediatric patients with SP, which is in line with our management. Noorbakhsh et al. and Fitzwater et al., who studied 183 and 96 children, respectively, concluded that short-term observation of a child with SP with mild symptoms in the emergency department might be sufficient [4,5]. In the study by Fitzwater et al., in most cases children were admitted to the hospital and some also to the ICU, although none of them had progression or were intubated [4]. We think that hospitalization for children suffering from SP secondary to viral infections might be reasonable to enable rest and monitoring, and especially to enable treatment and detection of imminent or already present complications.

In conclusion, SP related to viral infections appears to be a benign condition. Supportive treatment and rest are effective therapies for SP. Surveillance and monitoring might be reasonable to detect and treat potential complications.

## Figures and Tables

**Figure 1 children-09-01040-f001:**
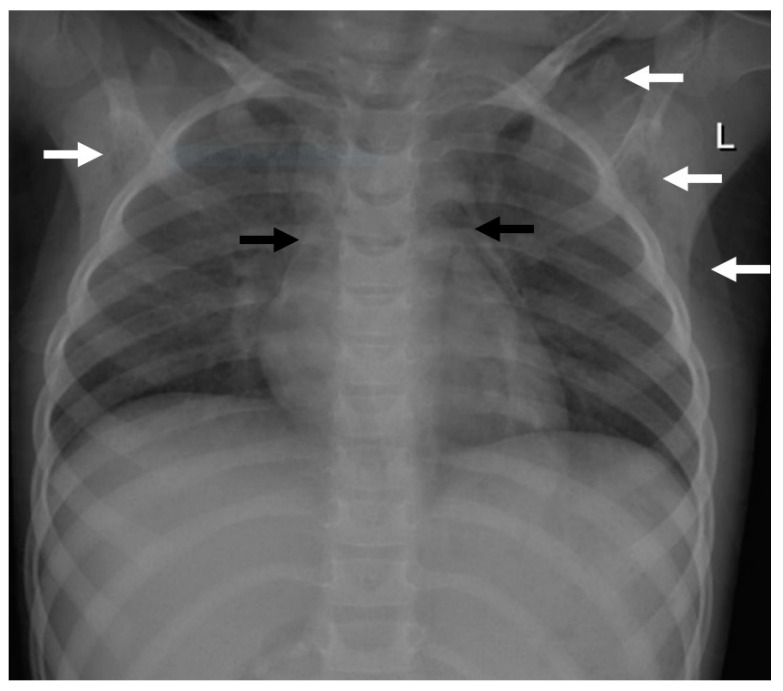
Imaging findings in patient 1. Chest X-ray showing a pneumomediastinum (black arrows) and a subcutaneous emphysema, more pronounced on the left site (white arrows).

**Figure 2 children-09-01040-f002:**
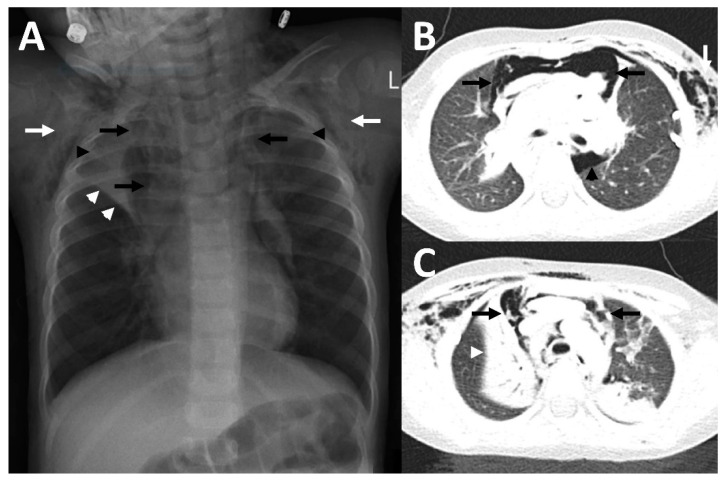
Imaging findings in patient 2. In the chest x-ray (**A**), as well as in CT-imaging (**B**,**C**) bilateral subcutaneous emphysema (white arrows) and a pneumomediastinum (black arrows) with a Spinnaker sail (angel wing) sign ((**A**), thymus lobes marked with black arrowheads) was detected. In addition, there were atelectases (white arrowheads), predominantly of the right upper lobe, as well as a pneumothorax (black arrowhead in (**B**)).

**Figure 3 children-09-01040-f003:**
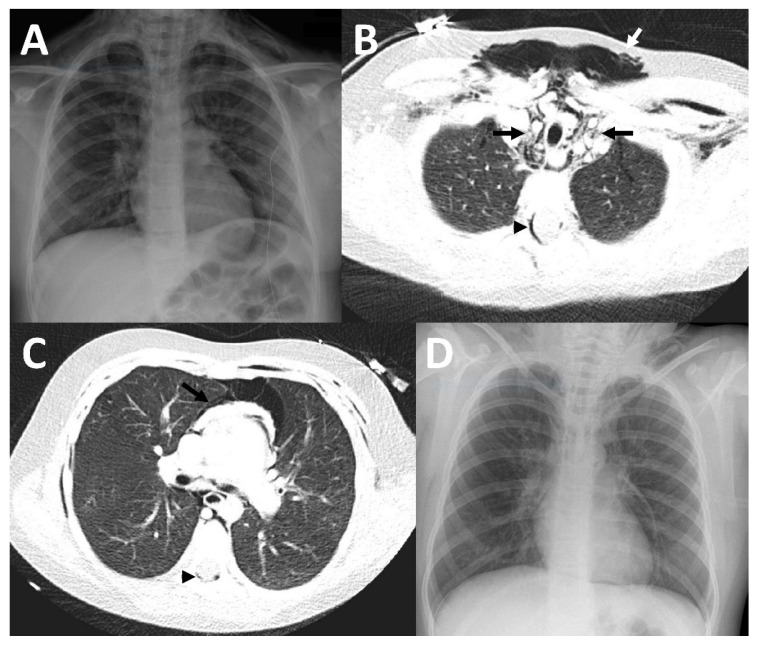
Imaging findings in patient 3. In the initial chest X-ray, pneumopericardium was suspected (**A**). Therefore, a CT was performed (**B**,**C**) which showed a pneumomediastinum (black arrows), accompanied by a large subcutaneous emphysema (white arrow) and epidural pneumatosis (black arrowhead), but no pneumopericardium. A chest X-ray (**D**) which was acquired at discharge showed substantial improvement of the previously mentioned findings.

## Data Availability

All relevant data are within the manuscript.

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
