# Peer review of "Spontaneous Pneumomediastinum in Children with Viral Infections: Report of Three Cases Related to Rhinovirus or Respiratory Syncytial Virus Infection"

_children, 2022, doi:10.3390/children9071040_

Round 1
Reviewer 1 Report
The paper by Leinert et al report three cases of SP, which were related to rhinovirus or respiratory syncytial virus infection. All three patients presented with typical symptoms of a respiratory tract infection and they were treated with oxygen supplementation, antibiotic therapy and supportive therapy and they were discharged after at least 7 days without remaining symptoms.
The paper is well written and it can be accepted with minor review.
Minor review
1. In all three cases the authors have to better explain antibiotic therapy (dose).
2. Why in the first case authors used ampicillin and in the second case ceftriaxone ev?
Author Response
We thank the reviewer for taking the time to read our manuscript and for his/her thoughtful comments.
1. The paper is well written and it can be accepted with minor review.
To 1. We thank the reviewer for this positive feedback.
Minor review
2. In all three cases the authors have to better explain antibiotic therapy (dose).
To 2. In the revised manuscript we have added the dose of the antibiotic therapy. Furthermore, the antibiotic agent for case 3 was also added.
3. Why in the first case authors used ampicillin and in the second case ceftriaxone iv?
To 3. In the first case ampicillin was given as it is a standard agent for community-acquired pneumonia. In the second case, ceftriaxone was already started by the referring hospital which transferred the patient to our hospital. Therefore, we proceeded with this therapy. We added this additional information in the revised manuscript.
Reviewer 2 Report
1) Abstract: L11-19. Spontaneous pneumomediastinum (SP) is generally a benign condition which can be caused by various etiologies. Data on SP related to respiratory viral infections in children is rare and there are currently no official guidelines or consistent treatment recommendations for these patients. We report three cases of SP, which were related to rhinovirus or respiratory syncytial virus (RSV) infection. All three patients presented with typical symptoms of a respiratory tract infection and required oxygen supplementation during the hospital stay. All children benefited from a con servative, supportive therapy, and bed rest, and could be discharged after at least 7 days without remaining symptoms. Surveillance and monitoring might be reasonable to detect and treat potential complications in children with SP due to viral infections, as one child developed an increasing pneumothorax, which had to be treated with a thoracic drainage. Please divide the abstract in different sections (background, aim, ..)
2) 1. Introduction L23-28 A pneumomediastinum is defined as the presence of interstitial air in the mediastinum. A spontaneous pneumomediastinum (SP) is caused by an increased pressure in the airways in comparison to the surrounding tissue, leading to a spread of air into the mediastinum. In this incidence, air is also pressed into the subcutaneous tissue or the pleural space, which is why SP is often accompanied with subcutaneous emphysema and pneumothoraces.[1]. Could you please improve this paragraph and add these references:
a- Radiological-pathological signatures of patients with COVID-19-related pneumomediastinum: is there a role for the Sonic hedgehog and Wnt5a pathways?. ERJ Open Res. 2021;7(3):00346-2021. doi:10.1183/23120541.00346-2021
b-Young convalescent COVID-19 pneumonia with extensive pneumomediastinum emphysema: Case report. Clin Case Rep. 2022;10(3):e05543. doi:10.1002/ccr3.5543
3) Introduction. L40-41. In this work, we report three cases of SP, which were related to rhinovirus or respiratory syncytial virus (RSV) infection in children and discuss treatment options for these patients, considering the recommendations for SP caused by different etiologies. Please underline the novelty of the study.
4) Figure 1. Imaging findings in patient 1. Chest x-ray showing a pneumomediastinum (black arrows) 57 and a subcutaneous emphysema, more pronounced on the left site (white arrows). Please add the Ct images of patients.
5) 4. Discussion L108-112. RSV and rhinovirus are two of the most common pathogens causing viral respiratory infections in children and are often seen in winter and beginning of spring.[9] Children under the age of 2 years, especially premature born toddlers, and those with chronic illnesses, such as immunodeficiencies or broncho-pulmonary disease are often hospitalized with RSV.[9]. Please summarise here the most important observations of the study.
6) 4. Discussion L150-152. In conclusion, SP related to viral infections appears to be a benign condition. Supportive treatment and rest are effective therapies for SP. Surveillance and monitoring might be reasonable to detect and treat potential complications.
Author Response
We thank the reviewer for taking the time to read our manuscript and for his/her thoughtful comments.
1. Abstract: L11-19. Please divide the abstract in different sections (background, aim, ...)
To 1. According to the suggestion of the reviewer, we have divided the abstract in different sections in the revised manuscript. When writing the first draft, we followed the “Instructions for Authors”, which states: "The abstract should be a single paragraph and should follow the style of structured abstracts, but without headings...". Therefore, we did not include the headings in the original manuscript.
2. Introduction L23-28: Could you please improve this paragraph and add these references:
a- Radiological-pathological signatures of patients with COVID-19-related pneumomediastinum: is there a role for the Sonic hedgehog and Wnt5a pathways?. ERJ Open Res. 2021;7(3):00346-2021. doi:10.1183/23120541.00346-2021
b-Young convalescent COVID-19 pneumonia with extensive pneumomediastinum emphysema: Case report. Clin Case Rep. 2022;10(3):e05543. doi:10.1002/ccr3.5543
To 2. We revised the paragraph as recommended by the reviewer. Additionally, we added the suggested references to the following section in which we address the occurrence of SP in COVID 19.
3. Introduction. L40-41: Please underline the novelty of the study.
To 3. In the previous section we explain the limited data on SP in children with respiratory infections: “SP related to respiratory viral infections in children is seldom and only a few case reports are published.[9,10] Studies focusing on SP due to respiratory viral infections are not available. Accordingly, data regarding the clinical course in these patients is rare, and there are currently no official guidelines or consistent treatment recommendations for those patients.” We hope that the reviewer agrees, that this section addresses the novelty of our report.
4. Figure 1. Imaging findings in patient 1. Chest x-ray showing a pneumomediastinum (black arrows) 57 and a subcutaneous emphysema, more pronounced on the left site (white arrows). Please add the Ct images of patients.
To 4. We agree, that an image of a CT scan would be helpful. Unfortunately, no CT scan was performed in this patient.
5. Discussion L108-112. Please summarise here the most important observations of the study.
To 5. According to the reviewers suggestion, we have Adresse the main observations in the revised manuscript.
6. Discussion L150-152. In conclusion, SP related to viral infections appears to be a benign condition. Supportive treatment and rest are effective therapies for SP. Surveillance and monitoring might be reasonable to detect and treat potential complications.
To 6. As no recommendation or demands for revision are stated by the reviewer regarding this section, we carefully perused the conclusion but did not find any needs for revision.